# CURING THE TRANSITIVITY CURSE: SHORTCUT LOGICAL REASONING VIA A PRIORI KNOWLEDGE COMPILATION

## ABSTRACT

While large language models (LLMs) have shown remarkable reasoning abilities, they often fail at multi-hop logical reasoning tasks that require chaining inferences, struggling to deduce transitive relations like $(P \rightarrow R)$ from $(P \rightarrow Q) \wedge (Q \rightarrow R)$. This fundamental limitation, which we term the **"Transitivity Curse"**, leads to brittle reasoning chains and significant error propagation. Existing reasoning frameworks, often based on Chain-of-Thought, attempt to traverse these long paths sequentially, a process that is both inefficient and prone to failure as complexity increases. To cure this curse, we introduce a novel mechanism designed to be integrated into existing logical reasoners. Our mechanism shifts the paradigm from passively traversing reasoning chains to proactively compiling them through a process we call **A Priori Knowledge Compilation (APKC)**. This process unfolds in two critical phases. First, it employs a goal-oriented backward analysis to identify a focused, relevant subgraph of the knowledge base. Subsequently, within this constrained boundary, our mechanism performs a systematic forward-chaining process to synthesize new knowledge in the form of both foundational **derived facts** and powerful **composite rules**. This compiled knowledge collapses multi-step inferences into fewer, more robust steps. By allowing a host framework to leverage this compiled knowledge, our mechanism enables a more direct form of **Shortcut Reasoning**, drastically reducing the required depth of runtime inference. Experiments show that when integrated into state-of-the-art reasoning frameworks, our mechanism consistently and significantly boosts their performance on several logical reasoning benchmarks. Our findings demonstrate that APKC, as a plug-in mechanism, is a critical component for making existing LLM-based reasoners more robust, efficient, and trustworthy.

## 1 INTRODUCTION

The advent of Large Language Models (LLMs), spurred by Chain-of-Thought (CoT) prompting (Wei et al., 2022b), has significantly advanced automated reasoning capabilities. This breakthrough inspired advanced strategies like Tree-of-Thought (Yao et al., 2023b) and Graph-of-Thought (), which emulate human cognitive patterns by exploring complex problem spaces step-by-step. However, while these methods excel at tasks requiring broad, heuristic exploration, they exhibit a critical fragility when confronted with the stringent demands of formal logical reasoning. This weakness is starkly exposed in their struggle with multi-hop transitive inferences—the fundamental challenge of reliably deducing a conclusion $R$ from a fact $P$ through a chain of rules such as $(P \rightarrow Q)$ and $(Q \rightarrow R)$. We term this fundamental limitation the **"Transitivity Curse"**: the propensity of LLMs to lose inferential coherence and propagate errors across long logical chains. To illustrate, consider the deductive problem shown in Figure 1. Given the fact *IsAustin*(the city, True) and the necessary rules—*IsAustin*($x, True) \rightarrow IsInTexas($x, True) and *IsInTexas*($x, True) \rightarrow IsInUSA($x, True)—a model may still fail to conclude *IsInUSA*(the city, True). Existing methods, acting as *path followers*, attempt to traverse these steps sequentially; however, they often fail mid-chain, culminating in aborted reasoning paths or factually unsound judgments, even if all preceding steps in the chain were logically sound. This reliance on a fragile, step-by-step traversal strategy

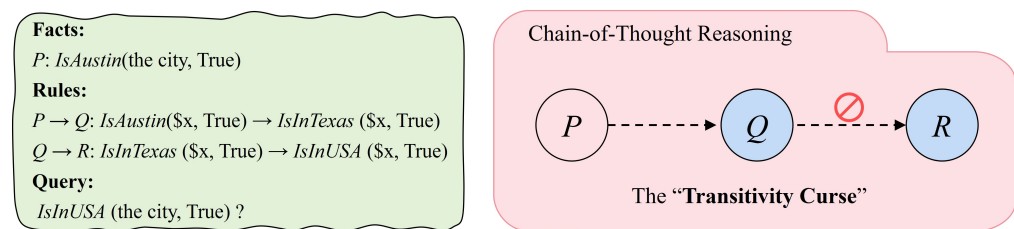

Figure 1: An illustration of the Transitivity Curse in Chain-of-Though Reasoning. Existing "path follower" methods attempt to sequentially traverse a fragile reasoning chain, where each step (e.g., $Q \rightarrow R$) is a potential point of failure.

exposes a critical gap in current methodologies: they lack a mechanism to proactively simplify or shorten the reasoning chain before execution.

Attempts to address the challenges of multi-hop reasoning can be broadly categorized into two dominant paradigms, neither of which is sufficient to cure the Transitivity Curse. The first is a neuro-symbolic approach that relegates the LLM to a mere translator for an external symbolic solver (**??**). While leveraging the solver's precision, this paradigm sidesteps the core problem of error propagation in long chains by simply offloading the reasoning task, failing to enhance the LLM's intrinsic capabilities. The second, a more integrated paradigm, uses symbolic Chain-of-Thought to perform reasoning entirely within the LLM (). Although this improves step-by-step faithfulness, it still meticulously traces the full logical path, thus remaining fundamentally vulnerable to the inefficiencies and error accumulation that define the curse. Ultimately, both paradigms remain fundamentally reactive *path followers*. They lack the ability to proactively restructure the problem by compiling logical steps into efficient shortcuts. This highlights the need for a new mechanism that does not just follow paths, but intelligently and reliably creates them.

To answer this call, we introduce a novel mechanism that operates as a true and effective *path creator*. The core of our approach is a process we term **A Priori Knowledge Compilation (APKC)**, a structured, two-phase procedure. First, it performs a goal-oriented backward analysis, starting from the query, to strategically prune the vast reasoning space down to a small, relevant subgraph of rules. With this focused scope defined, the mechanism then initiates a systematic forward-chaining process to synthesize new knowledge in the form of both foundational **derived facts** and powerful **composite rules**. This compiled knowledge fundamentally alters the reasoning landscape for the LLM. Its final deductive task is dramatically simplified, as it can now leverage the derived facts as new, reliable starting points and apply the composite rules to traverse what were previously long logical distances in fewer, more robust steps. This method directly confronts the Transitivity Curse by shortening the inferential chain required at runtime, with the set of all generated knowledge serving as a transparent and interpretable audit trail of the compilation process.

We empirically validate the effectiveness of our **APKC** mechanism through extensive evaluation on several challenging logical reasoning benchmarks. Our experiments demonstrate that integrating APKC into existing reasoners leads to significant performance gains over leading frameworks, particularly in scenarios requiring long-chain, transitive inferences. The main contributions of this work are threefold:

- We identify and characterize the "Transitivity Curse" as a key failure mode for LLM reasoners and, in response, propose a new reasoning paradigm that shifts from passive *path following* to proactive *path creation*.

- We design and implement the **APKC mechanism**, a concrete realization of this paradigm. It operates in two phases: (1) a goal-oriented backward analysis to prune the logical space, and (2) a constrained forward-chaining process to synthesize both foundational derived facts and powerful composite rules.

- We empirically validate that our APKC mechanism substantially improves both accuracy and efficiency on multi-hop reasoning tasks, while enhancing interpretability by producing an explicit, human-verifiable audit trail of the compiled knowledge.

## 2 RELATED WORK

**Sequential and Search-Based Reasoning Frameworks.** Recent achievements in reasoning research powered by LLMs have shown promising results Huang & Chang (2023); Dunivin (2024), significantly advancing their capabilities. The Chain-of-Thought (CoT) methodology Wei et al. (2022a) and its variants have been central to this progress, emulating human-like sequential reasoning. To overcome the limitations of linear paths, more advanced frameworks have introduced non-linear exploration, such as Tree-of-Thought Yao et al. (2023a) and Graph-of-Thought Besta et al. (2023); Zheng et al. (2024). These methods represent the state-of-the-art in what we term *path following*: they are sophisticated strategies for exploring and validating steps along a given reasoning chain.

**The Role of Symbolic Representation.** However, a parallel line of research has highlighted that the representational format of these reasoning steps is crucial. While natural language is versatile, studies show that structured representations can significantly bolster reasoning in specific domains. For instance, using pseudo-code for code generation tasks Li et al. (2023) or mathematical equations for math problems **?** leads to more robust outcomes. For formal logical reasoning, this is even more critical. The ambiguity of natural language is a primary contributor to the error propagation seen in the **Transitivity Curse**, where the fidelity of a long reasoning chain degrades with each step.

**Neuro-Symbolic Approaches for Logical Reasoning.** To bring more rigor to logical reasoning, one dominant neuro-symbolic paradigm uses LLMs as translators. Approaches like Logic-LM Pan et al. (2023) and LINC Olausson et al. (2023) leverage LLMs to convert natural language problems into formal languages like First-Order Logic, which are then passed to external symbolic solvers. While these methods benefit from the precision of classical solvers, they fail to enhance the LLM's intrinsic reasoning ability and merely offload the challenge of traversing a long deductive path. A different, more integrated approach might use symbols within a CoT framework, but it would still be meticulously tracing each step of the chain, thus remaining vulnerable to the curse.

In this work, we propose a fundamentally different approach. Instead of refining the process of *path following* (whether with better search or more robust symbolic steps), our work introduces a mechanism for proactive *path creation*. Our core contribution, **A Priori Knowledge Compilation** **(APKC)**, is a process that analyzes the symbolic structure of a problem before runtime deduction. It systematically synthesizes foundational facts and composite rules to build high-speed "shortcuts" across the knowledge graph. This compilation step directly confronts the Transitivity Curse by reducing the number of sequential steps required for a conclusion—a problem that both advanced search strategies and translator-based neuro-symbolic methods leave unaddressed.

## 3 PRELIMINARIES

**Logical Reasoning.** The task of multi-hop logical reasoning is to determine the truth value of a given hypothesis $H$ (e.g., True, False, Unknown) based on a knowledge base $\mathcal{K} = (\mathcal{F}, \mathcal{R}, \mathcal{P})$. This knowledge base consists of a set of known **Facts**, $\mathcal{F} = \{f_i\}$, and a set of **Rules**, $\mathcal{R} = \{r_i\}$. Each fact $f_i$ is a ground atomic formula representing a true statement (e.g., '*IsAustin*(the city, True)'). An atomic formula is composed of a **Predicate** $\mathcal{P}$, which describes a property or relation (e.g., *IsAustin*), and one or more arguments representing entities (e.g., 'the city'). Each rule $r_i$ is a conditional statement of the form $P \rightarrow Q$ (eg., *IsAustin*($x, True) \rightarrow IsInTexas($x, True)$), where the antecedent $P$ is a logical expression over atomic formulas and the consequent $Q$ is the resulting conclusion.

**LLM-based Reasoning Frameworks.** Modern LLM-based reasoners have evolved significantly beyond simple Chain-of-Thought (Wei et al., 2022a). To handle the stringent demands of formal logic, state-of-the-art frameworks such as SymbCoT (Xu et al., 2024b) and Aristotle (Xu et al., 2024a) integrate symbolic representations and structured architectures (e.g., plan-then-solve or decompose-search-resolve). These systems represent the pinnacle of sophisticated *path followers*, designed to traverse complex logical chains with high fidelity. However, their fundamental limitation is that they still meticulously trace these long paths step-by-step, leaving them vulnerable to the Transitivity Curse. This shared characteristic makes them ideal host frameworks for our mechanism, a proactive *path creator* designed to augment their step-by-step reasoning by pre-compiling the very inference chains they are tasked to follow.

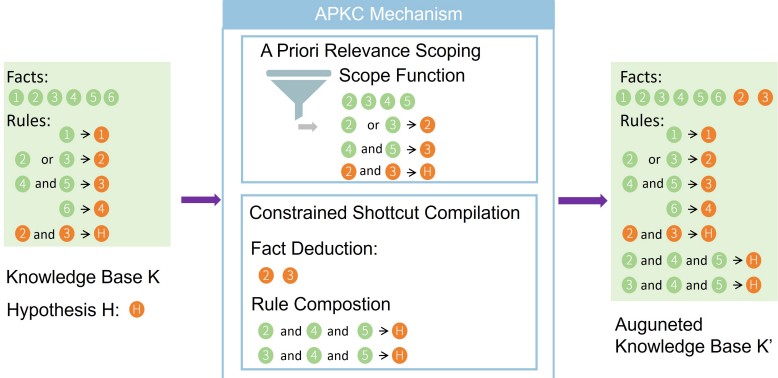

Figure 2: An overview of our **A Priori Knowledge Compilation (APKC)** mechanism. The mechanism operates in two distinct phases: (1) **A Priori Relevance Scoping** uses backward analysis from the query to prune the initial rule base to a small, relevant subset. (2) **Constrained Shortcut Compilation** then performs forward-chaining on this pruned set to synthesize new foundational facts and powerful composite rules. The final output is an augmented knowledge base that enables a downstream reasoner to perform more direct and robust Shortcut Reasoning.

## 4 METHODOLOGY

To cure the Transitivity Curse, we introduce our mechanism based on the principle of **A Priori Knowledge Compilation (APKC)**. This approach reframes the reasoning task from a reactive, step-by-step traversal to a proactive, two-phase procedure that first prunes the logical space and then compiles the necessary knowledge within it. Our mechanism operates on an initial knowledge base $\mathcal{K} = (\mathcal{F}, \mathcal{R})$ and a hypothesis $H$, with the objective of transforming $\mathcal{K}$ into an augmented knowledge base $\mathcal{K}' = (\mathcal{F}', \mathcal{R}')$. This new base contains synthesized derived facts and shortcut rules, drastically simplifying the final deductive task for a downstream reasoner.

As outlined in our introduction, the APKC mechanism is composed of two core phases, which we have termed: Phase 1: A Priori Relevance Scoping and Phase 2: Constrained Shortcut Compilation. In the following subsections, we provide a detailed operational description of each phase.

### 4.1 A PRIORI RELEVANCE SCOPING

This phase implements the 'Scope' function (Algorithm 1), which is the cornerstone of our *a priori* approach. Its primary objective is to mitigate the combinatorial explosion inherent in forward-chaining by identifying a small yet sufficient subset of rules, $\mathcal{R}_{\text{relevant}}$, pertinent to the hypothesis $H$, instead of reasoning over the entire rule base $\mathcal{R}$.

The 'Scope' function operates through a backward analysis that reasons about the relationships between rule **predicates**. We define a predicate as the symbolic relation in an atomic formula. The analysis begins by initializing a set of relevant predicates, $\mathcal{P}_{\text{relevant}}$, with all predicates found in the hypothesis $H$. It then iteratively expands this set by traversing the rule base $\mathcal{R}$ backwards: for any rule $r : P \to Q$ where the consequent's predicate is in $\mathcal{P}_{\text{relevant}}$,

---

**Algorithm 1** The A Priori Knowledge Compilation (APKC) Mechanism

**Require:** Initial knowledge base $\mathcal{K} = (\mathcal{F}, \mathcal{R}, \mathcal{P})$, Hypothesis $H$.

    *// — A Priori Relevance Scoping —*
1: $\mathcal{R}_{\text{relevant}} \leftarrow \text{Scope}(\mathcal{R}, \mathcal{P}, H)$

    *// — Constrained Knowledge Compilation —*
2: $\mathcal{F}_{\text{derived}} \leftarrow \text{DeduceFacts}(\mathcal{F}, \mathcal{R}_{\text{relevant}})$
3: $\mathcal{R}_{\text{shortcut}} \leftarrow \text{ComposeRules}(\mathcal{R}_{\text{relevant}})$

    *// — Return Augmented Knowledge Base —*
4: $\mathcal{F}' \leftarrow \mathcal{F} \cup \mathcal{F}_{\text{derived}}$
5: $\mathcal{R}' \leftarrow \mathcal{R} \cup \mathcal{R}_{\text{shortcut}}$
6: **return** $\mathcal{K}' = (\mathcal{F}', \mathcal{R}', \mathcal{P})$

all predicates from the antecedent $P$ are added to the set. This expansion continues until a fixed point is reached, ensuring all potentially relevant logical precursors are included.

With the set $\mathcal{P}_{\text{relevant}}$ fully constructed, we can now formally define the complete 'Scope' function as follows:

$$\text{Scope}(\mathcal{R}, \mathcal{P}, H) := \{r \in \mathcal{R} \mid \text{Predicates}(r) \subseteq \mathcal{P}_{\text{relevant}}\}$$

where $\text{Predicates}(r)$ is a function returning all predicates in rule $r$. The output of this function is the desired set of relevant rules, $\mathcal{R}_{\text{relevant}}$. This *a priori* pruning step dramatically reduces the search space, creating a constrained and computationally tractable environment for the compilation phase that follows.

### 4.2 CONSTRAINED SHORTCUT COMPILATION

With the constrained set of relevant rules, $\mathcal{R}_{\text{relevant}}$, produced by Phase 1, the mechanism now proceeds to the compilation phase. This phase is the core of the "path creator," where new knowledge is actively synthesized, corresponding to the 'DeduceFacts' and 'ComposeRules' functions in our main algorithm (Algorithm 1). The process involves two parallel operations: deriving all reachable facts and composing transitive rules into shortcuts.

The first operation is an exhaustive forward-chaining process to materialize all possible conclusions from the initial facts. This iterative process, encapsulated by the 'DeduceFacts' function, applies the rules in $\mathcal{R}_{\text{relevant}}$ to the initial facts $\mathcal{F}$ until a fixed point is reached. We formally define the function as computing the deductive closure over the initial facts, minus the initial facts themselves:

$$\text{DeduceFacts}(\mathcal{F}, \mathcal{R}_{\text{relevant}}) := \text{Closure}(\mathcal{F}, \mathcal{R}_{\text{relevant}}) - \mathcal{F}$$

where $\text{Closure}(\mathcal{F}, \mathcal{R}_{\text{relevant}})$ represents the complete set of facts derivable from $\mathcal{F}$ using the rules in $\mathcal{R}_{\text{relevant}}$. The output is the set of all newly derived facts, $\mathcal{F}_{\text{derived}}$.

Concurrently, the mechanism performs rule composition to create the shortcuts that directly cure the Transitivity Curse. The 'ComposeRules' function systematically identifies and synthesizes transitive rule pairs. We formally define this one-step composition as:

$$\text{ComposeRules}(\mathcal{R}_{\text{relevant}}) := \left\{ (P \to R) \,\middle|\, \begin{array}{l} \exists Q \text{ s.t. } (P \to Q) \in \mathcal{R}_{\text{relevant}} \\ (Q \to R) \in \mathcal{R}_{\text{relevant}} \end{array} \right\}$$

This process is repeated until no new shortcuts can be composed. The final output is the complete set of synthesized shortcut rules, $\mathcal{R}_{\text{shortcut}}$.

The outputs of this phase, $\mathcal{F}_{\text{derived}}$ and $\mathcal{R}_{\text{shortcut}}$, are then combined with the initial knowledge base to form the augmented knowledge base, $\mathcal{K}' = (\mathcal{F} \cup \mathcal{F}_{\text{derived}}, \mathcal{R} \cup \mathcal{R}_{\text{shortcut}})$. This enriched and computationally superior knowledge base is the final product of the APKC mechanism, ready to be passed to a downstream reasoner for a simplified and robust final judgment.

## 5 EXPERIMENTS

To evaluate the effectiveness of our **A Priori Knowledge Compilation (APKC)** mechanism, we formulate the following research questions (RQs):

**RQ1 (Overall Performance)** How effective is our APKC mechanism at improving the performance of state-of-the-art logical reasoners like SymbCoT and Aristotle?

**RQ2 (Scalability with Reasoning Depth)** How does the performance benefit of APKC scale as the logical depth of the reasoning tasks increase?

### 5.1 EXPERIMENTAL SETUP

**Models.** We assess the baselines and our method using GPT-4o-mini, Gemini-2.0-flash-lite, Gemma-3-27B-it and GPT-5. We also include GPT-5 to verify whether our method can generalize to Larger LLMs other than small LLMs.

| Model | Method | Dataset | | | | |
|-------|--------|----------------|--------|---------|-------------|----------|
| | | LogicalDeduction | AR-LSAT | PrOntoQA | ProofWriter | LogicNLI |
| **GPT-4o-mini** | CoT | 63.3 | 20.8 | 83.0 | 49.5 | 37.0 |
| | CR | 75.3 | 26.0 | 59.4 | 46.2 | 30.3 |
| | ToT | 72.3 | 23.8 | 59.8 | 48.3 | 29.7 |
| | DetermLR | 75.7 | 23.4 | 59.4 | 49.5 | 31.3 |
| | Aristotle | - | - | 62.4 | 61.9 | 49.2 |
| | **Aristotle + Ours** | **-** | **-** | **78.4** | **64.2** | **66.3** |
| | SymbCoT | 70.7 | 22.5 | 65.6 | 62.3 | 57.7 |
| | **SymbCoT + Ours** | **78.0** | **27.3** | **92.6** | **69.8** | **63.0** |
| **Gemini-2.0-flash-lite** | CoT | 71.3 | 19.1 | 87.6 | 56.7 | 39.8 |
| | CR | 76.3 | 30.7 | 82.2 | 62.3 | 37.7 |
| | ToT | 75.3 | 30.3 | 85.6 | 63.7 | 36.3 |
| | DetermLR | 75.7 | 29.9 | 78.2 | 60.7 | 35.0 |
| | Aristotle | - | - | 80.8 | 74.7 | 69.0 |
| | **Aristotle + Ours** | **-** | **-** | **84.4** | **81.9** | **69.7** |
| | SymbCoT | 78.3 | 25.5 | 99.8 | 76.7 | 63.7 |
| | **SymbCoT + Ours** | **89.0** | **31.2** | **99.8** | **84.2** | **65.7** |
| **Gemma-3-27B-it** | CoT | 78.0 | 30.3 | 99.4 | 67.5 | 38.3 |
| | CR | 83.3 | 31.2 | 93.2 | 71.2 | 38.0 |
| | ToT | 85.3 | 35.1 | 93.4 | 69.8 | 36.7 |
| | DetermLR | 86.0 | 30.7 | 92.0 | 71.2 | 35.3 |
| | Aristotle | - | - | 79.4 | 77.8 | 53.7 |
| | **Aristotle + Ours** | **-** | **-** | **94.4** | **79.0** | **60.0** |
| | SymbCoT | 73.7 | 37.7 | 99.0 | 81.5 | 66.7 |
| | **SymbCoT + Ours** | **80.3** | **41.1** | **99.6** | **83.8** | **67.3** |
| **GPT-5** | CoT | 83.8 | 40.3 | 100.0 | 98.0 | 47.3 |
| | CR | 90.1 | 84.9 | 98.8 | 97.6 | 89.1 |
| | ToT | 85.4 | 90.5 | 99.6 | 95.7 | 92.3 |
| | DetermLR | 84.6 | 86.8 | 99.8 | 96.9 | 93.3 |
| | Aristotle | - | - | 94.6 | 81.9 | 68.7 |
| | **Aristotle + Ours** | **-** | **-** | **99.8** | **92.0** | **77.0** |
| | SymbCoT | 98.3 | 98.3 | 100.0 | 94.2 | 95.0 |
| | **SymbCoT + Ours** | **99.7** | **98.3** | **100.0** | **94.3** | **97.3** |

Table 1: Proof accuracy of different methods across five logical reasoning datasets on GPT-4o-mini, Gemini-2.0-flash-lite, Gemma-3-27B-it and GPT-5.

**Datasets.** To verify the capability of LLMs to engage in rigorous logical reasoning based solely on established conditions, without external knowledge, we carefully select five challenging logical reasoning benchmarks: (1) **PrOntoQA** is similar to ProofWriter for evaluating logical reasoning. (2) **ProofWriter** is a widely used logical reasoning benchmark. We use the open-world assumption subset where each case requires to be proven true, false or unknown. We use the depth-5 subset containing 600 cases for evaluation. (3) **LogicNLI** is a challenging benchmark requiring complex first-order logic reasoning to solve. We follow the official data split and choose the validation set containing 300 examples for evaluation. (4) **LogicalDeduction (LD)** is a challenging task in Big-Bench. The problems are mainly about deducing the order of objects from a set of conditions. We use the full test set containing 300 examples for evaluation. (5) **AR-LSAT** is a challenging task in BigBench (). The problems are mainly about deducing the order of objects from a set of conditions. We use the full test set containing 300 examples for evaluation.

**Baselines.** We compare with a wide range of established baselines. Those baselines can be classi-fied into three main categories. (1) **Linear Reasoning** (LR) refers to approaches where the model arrives at an answer through a single-step process, using a straightforward response based on the ini-tial prompt including: *CoT* ; (2) **Aggregative Reasoning** (AR) refers to methods where the model performs reasoning multiple times or aggregates the results to reach a final answer. This includes: *Cumulative Reasoning* (CR; ); *DetermLR* ; *ToT* ; (3) **Symbolic Reasoning** (SR), which engages symbolic expressions and rules in the reasoning framework including: *Aristotle* and *SymbCoT* .

## 5.2 OVERALL PERFORMANCE (RQ1)

The main results of our evaluation are presented in Table 1. The findings strongly support the effectiveness of our APKC mechanism, leading to the following key observations:

**APKC consistently enhances state-of-the-art reasoners.** Our mechanism is not a standalone reasoner but a plug-in component. The results clearly show that integrating APKC provides a significant performance boost to already powerful host frameworks. For instance, on GPT-4o-mini, augmenting SymbCoT with APKC improves accuracy on LogicalDeduction from 70.7% to 78.0%, and on ProofWriter from 62.3% to 69.8%. Similarly, enhancing Aristotle raises its performance on LogicNLI from 49.2% to a much more competitive 66.3%. This demonstrates that APKC effectively serves as a powerful enhancement, enabling SOTA reasoners to reach new performance levels.

**The advantage of APKC is most pronounced on complex, deep-reasoning tasks.** This trend directly validates our core thesis. On simpler datasets like PrOntoQA, where baselines already approach a performance ceiling (e.g., 99.8% for SymbCoT on Gemini), the room for improvement is minimal. However, on more challenging datasets that feature longer and more complex deductive paths, APKC's "path creation" ability shines. The most dramatic improvements are seen on LogicNLI, the most difficult dataset, where APKC boosts Aristotle's performance by a remarkable 17.1 points on GPT-4o-mini (from 49.2% to 66.3%) and by 6.3 points on Gemma-3-27B-it (from 53.7% to 60.0%). This shows that APKC is not just a minor refinement but a crucial tool for curing the Transitivity Curse where it is most severe.

**Our mechanism is generalizable across different models.** Our results confirm that APKC's benefits are not confined to a specific model or architecture. **First**, it is architecturally general, successfully enhancing both the planner-solver approach of SymbCoT and the decomposer-searcher approach of Aristotle. **Second**, it is scalable across different model sizes. For smaller models like GPT-4o-mini and Gemma-3-27B-it, APKC provides a critical boost, often elevating their performance to the level of much larger models. For the most powerful models like GPT-5, where baselines are already extremely high, APKC still provides a consistent edge, pushing the frontier of performance even further (e.g., improving SymbCoT on LogicNLI from 95.0% to 97.3%).

## 5.3 SCALABILITY WITH REASONING DEPTH (RQ2)

Having established the overall effectiveness of our APKC mechanism, we now turn to a more fine-grained analysis of its performance as a function of reasoning depth. This RQ is designed to directly test our central hypothesis: that APKC's primary advantage lies in its ability to cure the Transitivity Curse, a problem that becomes exponentially more severe as the length of the inferential chain increases.

The results, visualized in Figure 3, provide a clear and compelling validation of this hypothesis. First, we observe a textbook illustration of the Transitivity Curse in the vanilla CoT baseline: its accuracy plummets from 80.3% at depth-0 to a mere 52.6% at depth-5, demonstrating a catastrophic decay in performance as the path lengthens. The more robust SymbCoT framework, while significantly better, is not immune; its performance also steadily degrades from 95.0% down to 76.3%. This shows that even a

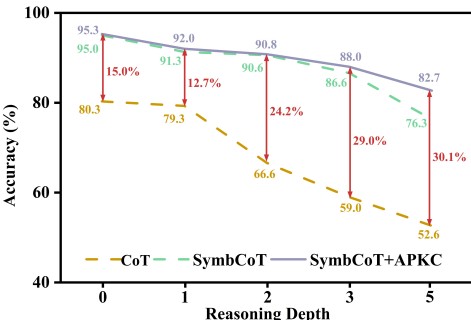

Figure 3: On the ProofWriter dataset (using Gemini-2.0-flash-lite), our APKC mechanism's performance advantage over the CoT baseline widens as reasoning depth increases, a gap highlighted by the red dual-headed arrows.

sophisticated *path follower* eventually succumbs to the challenges of a long journey.

The true impact of our *path creator* is revealed when we analyze the performance delta between SymbcoT+APKC and SymbCoT. At shallow depths (0-2), where SymbCoT is already highly effective, our APKC provides a modest but consistent improvement. However, as the reasoning depth increases, the advantage conferred by APKC becomes dramatically more pronounced. The perfor-

mance gap, which is marginal at depth-1 (+0.7%), widens to +1.4% at depth-3, and culminates in a substantial +6.4% advantage at a reasoning depth of 5. This trend powerfully demonstrates that APKC is not a mere incremental improvement; its value becomes most critical precisely when the Transitivity Curse is at its most potent. By pre-compiling the long inferential chain into shorter, more robust steps, our mechanism provides a resilient solution that scales effectively to complex, multi-hop reasoning challenges.

## 6  CONCLUSION

In this paper, we identified and characterized the "Transitivity Curse" as a fundamental vulnerability in modern Large Language Models, where the reliability of reasoning decays rapidly with the length of the inferential chain. We argued that existing state-of-the-art reasoning frameworks, while sophisticated, operate as reactive *path followers*, leaving them susceptible to this curse. To address this critical gap, we introduced **A Priori Knowledge Compilation (APKC)**, a novel plug-in mechanism that transforms these models into proactive *path creators*. By employing a two-phase process of goal-oriented relevance scoping and constrained forward-chaining, APKC proactively synthesizes foundational derived facts and powerful composite rules. This compiled knowledge effectively shortens the deductive paths that the host reasoner must traverse at runtime. Our extensive experiments empirically validated that integrating APKC significantly enhances the performance of leading logical reasoning frameworks like SymbCoT and Aristotle. Crucially, we demonstrated that the performance advantage of our mechanism becomes more pronounced as the reasoning depth increases, providing direct evidence that it successfully mitigates the Transitivity Curse. Our work suggests that for LLMs to achieve robust, high-fidelity reasoning, the focus must expand from merely improving path-traversal strategies to also include proactive, intelligent knowledge preparation. Future work could explore the scalability of this compilation approach to vastly larger knowledge bases and its applicability to other reasoning domains such as planning and causal inference.

## 7  STATEMENT ON THE USE OF LARGE LANGUAGE MODELS (LLMS)

In the preparation of this manuscript, Large Language Models (LLMs) were utilized as a general-purpose writing assistance tool. The specific roles of the LLMs were confined to the following aspects:

**Language Polishing:** Improving sentence structure, diction, and grammar to enhance the clarity and readability of the text.

**Grammar and Spelling Correction:** Assisting in proofreading the manuscript to correct potential grammatical errors and spelling mistakes.

It is important to state that all core research ideas, experimental design, data analysis, interpretation of results, and the final conclusions presented in this paper are entirely the original work of the authors. The LLMs did not contribute to any of the conceptual or substantive aspects of the research, such as ideation, methodology design, or drawing conclusions. Their use was strictly limited to improving the linguistic quality of the manuscript. Therefore, the LLMs do not qualify as authors or contributors to this work.

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
