# OpenReview forum: "Curing the Transitivity Curse: Shortcut Logical Reasoning via A Priori Knowledge Compilation"
_ICLR.cc/2026/Conference — ICLR 2026 Conference Withdrawn Submission_

### Official Review · Reviewer_Zciy · 2025-10-23

**Soundness:** 2
**Presentation:** 2
**Contribution:** 2
**Rating:** 4
**Confidence:** 3

**Summary:**

The paper identifies the “Transitivity Curse”, referring to the difficulty that large language models (LLMs) face when deducing transitive relations in logical reasoning tasks. To address this, the authors propose APKC, a two-phase approach designed to reduce the search depth of logical reasoning problems. Experimental results demonstrate that APKC improves upon state-of-the-art logical reasoning baselines.

**Strengths:**

- The idea is interesting

- Paper is generally clear and easy to follow.

**Weaknesses:**

- While the motivation and core idea are clear, the writing could be further refined for clarity and emphasis. The key contribution to me is reducing the search space in logical reasoning tasks to achieve conclusions with less computation. I would recommend to state it more explicitly.

- The methodology requires additional clarification. In particular, Sections 4.1 and 4.2 are ambiguous: are these steps performed using LLMs, or implemented through another mechanism?

- Deeper analysis and discussion would enhance the paper, particularly regarding the implications and limitations of the approach.

- Typo:
Line 42: Graph-of-Thought lacks a citation.
Lines 71–72, 75, and 315: citations are also missing or inconsistent.

**Questions:**

- Are the proposed methods compatible with all reasoning baselines?

- Specifically, can they be integrated with CoT or ToT approaches?

---

### Official Review · Reviewer_aWyS · 2025-11-03

**Soundness:** 1
**Presentation:** 1
**Contribution:** 2
**Rating:** 2
**Confidence:** 3

**Summary:**

This paper introduces "Transitivity Curse", which refers to the hypothesis that LLMs struggle to deduce transitive relations like "from $(P \rightarrow Q) \wedge (Q \rightarrow R)$ inferring $(P \rightarrow R)$". Then it proposes "A Priori Knowledge Compilation" (APKC), a process that employs (1) a goal-oriented backward analysis for subgraph identification of a knowledge base and (2) systematic forward-chaining to synthesize new knowledge. It enhances LLMs' reasoning abilities and improves their performance on several logical reasoning benchmarks.

**Strengths:**

- Logical reasoning limitation is an important problem for LLMs both in theory and in practice, and this work aims to improve LLM logical reasoning.
- The empirical evaluations of this paper are done with several different models and datasets, indicating generality and external validity.
- The work aims to go beyond performance by analyzing scaling behaviors with reasoning depth.

**Weaknesses:**

- The biggest weakness of this paper is about the notion of Transitivity Curse. The authors claim that LLMs have this problem but do not cite any concrete evidence. Nor do the authors carry out any targeted experiments to confirm the Transitivity Curse hypothesis. As a result, the premise of this work is unsound.
- I find the writing quality of this work unsatisfactory. There are many missing cites in the paper, represented by empty parentheses and (??). The writing significantly lacks precision. For example, methods like Tree-of-Thoughts (the paper incorrectly uses the singular form) are no longer "state-of-the-art" (Sec. 2). In general, many of the sentences seem to lack coherence.
- Much work has done in the space of LLM and logical reasoning. The Related Work section (Sec. 2) is really incomplete. In fact no single paper from 2025 is cited---this is not acceptable given how much work comes out this year on language models and (logical) reasoning.

**Questions:**

See weaknesses.

---

### Official Review · Reviewer_ZEoX · 2025-11-08

**Soundness:** 2
**Presentation:** 2
**Contribution:** 2
**Rating:** 2
**Confidence:** 4

**Summary:**

This paper addresses a weakness in LLMs called the transitivity curse, the tendency for LLMs to fail at multi-hop logical reasoning tasks that require chaining inferences (e.g., deducing P→R from P→Q and Q→R). The authors propose A Priori Knowledge Compilation (APKC), a plug-in mechanism that transforms reasoning from passive "path following" to proactive "path creation."by operateing in two phases:
1: Priori Relevance Scoping: Uses backward analysis from the query to identify relevant rules, pruning the reasoning space 2: Constrained Shortcut Compilation: Performs forward-chaining to synthesize new derived facts and composite rules that create "shortcuts" across the knowledge graph.

**Strengths:**

The method works as a plug-in for different reasoning frameworks (SymbCoT and Aristotle), demonstrating architectural flexibility.

**Weaknesses:**

1. **Insufficient Literature Review and Positioning**:
   - The paper lacks comprehensive coverage of recent advances in LLM reasoning. While the term "Transitivity Curse" may be novel, the underlying challenge is well-documented in recent literature on proactive reasoning, multi-hop inference, and long-horizon reasoning tasks, none of which are adequately discussed.
   - The related work section and baseline comparisons are limited to CoT and its variants (Tree-of-Thought, Graph-of-Thought), which represent somewhat dated approaches from 2022-2023. The paper fails to engage with more recent reasoning frameworks and techniques.
   - Despite mentioning neuro-symbolic methods and external symbolic solvers (e.g., Logic-LM, LINC), the paper provides no empirical comparison with these approaches, leaving a critical gap in understanding how APKC compares to existing solutions that also address multi-hop reasoning challenges.

2. **Limited Novelty and Scope**:
   - The A Priori Knowledge Compilation approach bears strong resemblance to existing work in agentic memory systems and RAG (Retrieval-Augmented Generation) frameworks, which similarly involve pre-computing knowledge structures and rules for improved reasoning. The paper fails to differentiate its contribution from these established approaches.
   - The method relies entirely on prompting techniques without exploring how post-training approaches (e.g., fine-tuning, RLHF, or specialized training objectives) could enhance the reasoning capabilities. This represents a missed opportunity to demonstrate deeper integration with model training.
   - The evaluation is restricted to formal logical reasoning benchmarks, leaving questions about the method's applicability to real-world reasoning tasks, commonsense reasoning, or other domains where transitivity and multi-hop inference are important.
   - These limitations collectively diminish the paper's novelty and impact, as it appears to be an incremental application of known techniques rather than a fundamental advance in addressing reasoning challenges.

**Questions:**

1. How does APKC scale with knowledge base size? The paper shows results on small benchmarks (300-600 examples), but real-world knowledge bases can contain millions of facts and rules. What is the computational complexity of the exhaustive forward-chaining in Phase 2, and at what point does the compilation become intractable? Please provide runtime analysis and experiments on larger knowledge bases.

2. If LLMs struggle with multi-hop transitive reasoning (the "Transitivity Curse"), why should we trust the same LLMs to correctly perform the compilation steps that require similar transitive reasoning? Isn't this circular - using an LLM's reasoning to fix its own reasoning limitations? How do you ensure the compilation process doesn't introduce or propagate errors?

3. How does APKC differ from classical forward-chaining algorithms or modern RAG/memory systems that similarly pre-compute and cache derived knowledge?

Also, multiple citation formatting errors appear throughout the paper (lines 42, 71, 123) please double check it.

---

### Official Review · Reviewer_uoXY · 2025-11-20

**Soundness:** 2
**Presentation:** 2
**Contribution:** 2
**Rating:** 4
**Confidence:** 3

**Summary:**

In this paper, the authors point out that large language models suffer from the "Transitivity Curse" in multi-hop logical reasoning—they struggle to deduce P→R from (P→Q)∧(Q→R) and are prone to error accumulation. To address this, it proposes the A Priori Knowledge Compilation (APKC) mechanism. APKC first filters relevant rules through goal-oriented backward analysis, then synthesizes derived facts and composite rules via constrained forward chaining to shorten reasoning chains. Experiments show that when integrated as a plug-in into mainstream reasoning frameworks like SymbCoT, APKC significantly improves performance on five logical reasoning benchmark datasets. The advantage becomes more pronounced as reasoning depth increases, and it is compatible with models of different scales, effectively alleviating the Transitivity Curse.

**Strengths:**

1. In general, the idea is interesting and possibly useful.
2. The authors introduced a new logic reasoning paradigm.

**Weaknesses:**

1. Lacks of experiments to support the main motivation: the authors strive to solve the problem of Revision Curse, yet this paper lacks of comparisons with the reasoning-related work of current state-of-the-art approaches, and the comparative results with some existing reasoning methods are also unconvincing.
2. Unclear and incomplete paper writing: there is a lot of places missing appropriate citations, and the quality of the tables is also not very good.

**Questions:**

Please see the weakness part.

---

### Note · Authors · 2026-01-13

**Comment:**

We have decided to withdraw this submission. We thank the reviewers for their time and feedback.

**Withdrawal Confirmation:**

I have read and agree with the venue's withdrawal policy on behalf of myself and my co-authors.